# (+)-Lipoic Acid Reduces Lipotoxicity and Regulates Mitochondrial Homeostasis and Energy Balance in an In Vitro Model of Liver Steatosis

**DOI:** 10.3390/ijms241914491

**Published:** 2023-09-23

**Authors:** Lucia Longhitano, Alfio Distefano, Angela Maria Amorini, Laura Orlando, Sebastiano Giallongo, Daniele Tibullo, Giuseppe Lazzarino, Anna Nicolosi, Amer M. Alanazi, Concetta Saoca, Vincenzo Macaione, M’hammed Aguennouz, Federico Salomone, Emanuela Tropea, Ignazio Alberto Barbagallo, Giovanni Li Volti, Giacomo Lazzarino

**Affiliations:** 1Department of Biomedical and Biotechnological Sciences, University of Catania, 95123 Catania, Italy; lucia.longhitano@unict.it (L.L.); distalfio@gmail.com (A.D.); amorini@unict.it (A.M.A.); lauraorlando2810@gmail.com (L.O.); sebastiano.giall@gmail.com (S.G.); d.tibullo@unict.it (D.T.); lazzarig@unict.it (G.L.); tropeaemanuela3@gmail.com (E.T.); ignazio.barbagallo@unict.it (I.A.B.); 2Hospital Pharmacy Unit, Ospedale Cannizzaro, 95125 Catania, Italy; annanicolosi@hotmail.com; 3Pharmaceutical Biotechnology Laboratory, Department of Pharmaceutical Chemistry, College of Pharmacy, King Saud University, Riyadh 11451, Saudi Arabia; amalanazi@ksu.edu.sa; 4Department Clinical and Experimental Medicine, University of Messina, Via Consolare Valeria 1, 98125 Messina, Italy; saoca.concetta@unime.it (C.S.); vincenzo.macaione@unime.it (V.M.); aguenoz.mhommed@unime.it (M.A.); 5Division of Gastroenterology, Ospedale di Acireale, Azienda Sanitaria Provinciale di Catania, 95024 Catania, Italy; federicosalomone@rocketmail.com; 6Departmental Faculty of Medicine and Surgery, UniCamillus-Saint Camillus International University of Health Sciences, Via di Sant’Alessandro 8, 00131 Rome, Italy; giacomo.lazzarino@unicamillus.org

**Keywords:** steatosis, lipoic acid, mitochondrial dynamics, metabolism

## Abstract

Non-alcoholic fatty liver disease (NAFLD) is characterized by the accumulation of lipids within hepatocytes, which compromises liver functionality following mitochondrial dysfunction and increased production of reactive oxygen species (ROS). Lipoic acid is one of the prosthetic groups of the pyruvate dehydrogenase complex also known for its ability to confer protection from oxidative damage because of its antioxidant properties. In this study, we aimed to investigate the effects of lipoic acid on lipotoxicity and mitochondrial dynamics in an in vitro model of liver steatosis. HepG2 cells were treated with palmitic acid and oleic acid (1:2) to induce steatosis, without and with 1 and 5 µM lipoic acid. Following treatments, cell proliferation and lipid droplets accumulation were evaluated. Mitochondrial functions were assessed through the evaluation of membrane potential, MitoTracker Red staining, expression of genes of the mitochondrial quality control, and analysis of energy metabolism by HPLC and Seahorse. We showed that lipoic acid treatment restored membrane potential to values comparable to control cells, as well as protected cells from mitochondrial fragmentation following PA:OA treatment. Furthermore, our data showed that lipoic acid was able to determine an increase in the expression of mitochondrial fusion genes and a decrease in mitochondrial fission genes, as well as to restore the bioenergetics of cells after treatment with palmitic acid and oleic acid. In conclusion, our data suggest that lipoic acid reduces lipotoxicity and improves mitochondrial functions in an in vitro model of steatosis, thus providing a potentially valuable pharmacological tool for NAFLD treatment.

## 1. Introduction

Alpha-lipoic acid or lipoic acid is a natural antioxidant found in dietary sources like red meat, spinach, and broccoli [1]. It is known for its fundamental role as the prosthetic group of the E2 subunit of the pyruvate dehydrogenase and alpha-ketoglutarate dehydrogenase complexes, both crucial for allowing adequate cell energy production. When lipoate is in its fully reduced form, it is ascribed to possess antioxidant effects [1,2,3] and potential therapeutic uses in various conditions, including liver steatosis [4,5]. Moreover, lipoic acid has been shown to enhance mitochondrial biogenesis [6,7], the process responsible for the formation of new mitochondria within cells. Animal and human studies have also demonstrated that lipoic acid increases the expression of genes involved in mitochondrial biogenesis and improves mitochondrial function [4,8,9,10,11]. As a result, lipoic acid has been investigated for its potential therapeutic benefits in metabolic disorders characterized by mitochondrial dysfunction [12,13]. However, the specific molecular mechanisms underlying lipoic acid’s beneficial effects in such pathological conditions remain elusive and require further investigation. 

Maintaining mitochondrial function and cellular homeostasis relies on mitochondrial dynamics, encompassing the delicate equilibrium of the processes of mitochondrial fusion and fission [14]. Fusion involves the merging of mitochondria, while fission involves their division into smaller units [15]. Both processes are tightly regulated, and disturbances can lead to mitochondrial dysfunction commonly encountered in different pathological conditions [15,16]. Interestingly, lipoic acid has been found to modulate mitochondrial fusion and fission by regulating key genes involved in these processes. Notably, lipoic acid increases the expression of mitochondrial fusion-related genes, such as mitofusin-1 and -2 (Mfn1 and 2), concomitantly inhibiting the expression of mitochondrial fission-related genes like dynamin-related protein 1 (Drp1) [17,18]. These effects on mitochondrial dynamics are believed to contribute to lipoic acid’s overall benefits on mitochondrial function and cellular homeostasis. By promoting fusion and inhibiting fission, lipoic acid contributes to the maintenance of mitochondrial integrity and prevents dysfunction, which is associated with various diseases, including neurodegenerative disorders and metabolic conditions like non-alcoholic fatty liver disease (NAFLD) [10,19].

NAFLD is a metabolic disorder characterized by the accumulation of fat in the liver [20]. Mitochondrial dysfunction has been implicated in its pathogenesis [21], and recent studies have suggested a connection between mitochondrial fusion, fission, and the development of NAFLD [22,23]. In NAFLD, there is an increase in mitochondrial fission and a decrease in fusion, resulting in a fragmented and dysfunctional mitochondrial network. This contributes to fat accumulation in the liver since dysfunctional mitochondria produce less ATP and are more susceptible to oxidative stress, leading to lipid buildup and inflammation [21,24].

The objective of this study is to evaluate the potential beneficial effects of lipoic acid in an in vitro model of liver steatosis, with a specific focus on mitochondrial dynamics and the cellular, mitochondrial-related metabolic profile.

## 2. Results

### 2.1. Lipoic Acid Restores the Effect of Palmitic Acid/Oleic Acid on HepG2 Cell Viability

We first evaluated the effect of PA:OA (palmitic acid/oleic acid) on HepG2 cell viability. Our results showed that PA:OA treatment caused a significant decrease in normalized cell index and area under the curve (AUC) by 1.8 times compared to control cells, indicating a decrease in cell viability (Figure 1A,B, Table 1). We then evaluated the effect of increasing doses of lipoic acid on HepG2 cells previously challenged with PA:OA. Our results (Figure 1A,B and Table 1) showed that both lipoic acid concentrations (1 and 5 µM) were able to counteract the effect of PA:OA, resulting in a significant increase in cell proliferation, as indicated by normalized cell index and AUC (*p* < 0.05 compared to PA:OA treated cells, not significantly different from control cells). However, the addition of lipoic acid, in the absence of PA:OA pretreatment, produced a significant increase in cell proliferation compared to untreated cells (Appendix A).

### 2.2. Lipoic Acid Improves Steatosis and Reduces the Area of Lipid Droplets in PA:OA-Treated HepG2 Cells

Given the effect of α-lipoic acid on cell proliferation of PA:OA-treated HepG2 cells, we analyzed the effect on steatosis and accumulation of lipid droplets, both after 24 h and 48 h of lipoic acid treatment. Our data show that PA:OA significantly increased the number of lipid droplets (64.4% and 82.7% of steatosis compared to control cells, *p* < 0.05) and that both doses of lipoic acid (1 and 5 μM) induced a significant reduction in the % of steatosis, either after 24 h (Figure 2A and Figure 3A, Table 2) or 48 h (Figure 2B and Figure 4A, Table 2) of treatment (*p* < 0.05 compared to HepG2 cells treated with PA:OA only).

Interestingly, we showed that within the steatosis-positive cell population, PA:OA significantly increased lipid droplet area (+12.8% and +12.2% of macro steatosis compared to control cells, *p* < 0.05) and that, also in this case, the percentage of macro steatosis was significantly reduced by both doses of lipoic acid at both time points after treatment (*p* < 0.05 compared to HepG2 cells treated with PA:OA only) (Figure 2C,D, Figure 4B and Figure 5B, Table 2).

### 2.3. Lipoic Acid Restores Mitochondrial Membrane Potential and Inhibits Fragmentation

As shown in Figure 5A, untreated HepG2 cells displayed large numbers of green, fluorescent mitochondria, representing J aggregates accumulating at the normally hyperpolarized membrane potential. Conversely, PA:OA-treated HepG2 cells had fewer red J aggregates, indicating a gradual dissipation of Δψm with a rapid and significant reduction in the percentage of polarized cells (AUC: 112.34 ± 39.6) (−77.6% AUC compared to control cells, *p* < 0.05) (Figure 5A,B). Treatment with lipoic acid (1 and 5 μM) produced a significant recovery of Δψm compared to PA:OA-treated cells (Figure 5A,B); in fact, as shown in Figure 5A,B, both the treatment with lipoic acid of 1 and 5 μM restored Δψm to values comparable to those measured in control naïve cells and significantly higher than those recorded in PA:OA-treated cells (+77.8% and +77.7% AUC, *p* < 0.05).

Given this effect, we then evaluated mitochondrial fragmentation using MitoTracker Red staining. Our data showed that PA:OA treatment alone caused a significant increase in mitochondrial fragmentation (+27.0-fold change over control, *p* < 0.05) and area under the curve (*p* < 0.05) (Figure 5C,D). Figure 5E shows representative images of mitochondria fragmentation in control cells and in cells following treatment with PA:OA (Figure 5E). Again, lipoic acid was able to determine a significant reduction in mitochondrial fragmentation compared to PA:OA-treated HepG2 cells (−45.8% and −64.3% AUC with 1 μM and 5 μM lipoic acid, respectively, *p* < 0.05) (Figure 5C–E), suggesting the general lipoic-acid-mediated improvement of mitochondrial function.

To further evaluate the beneficial effects of lipoic acid on mitochondrial function, we analyzed the expression of a panel of genes involved in mitochondrial dynamics and mitochondrial biogenesis. Treatment of HepG2 cells with PA:OA only induced a significant decrease in the relative mRNA expression levels of the MFN1 (Mitofusin 1) (0.44 ± 0.07 vs. 1.0 ± 0.2 in control) and MFN2 (Mitofusin 2) (0.56 ± 0.1 vs. 1.0 ± 0.03 in control) genes involved in the mitochondrial fusion process compared to control cells (Figure 6A,B) and a significant increase in the relative expression levels of mRNAs of the FIS1 (fission, Mitochondrial 1) (1.49 ± 0.25 vs. 1.0 ± 0.1 control) gene, which instead regulates mitochondrial fission (Figure 6C). Conversely, in HepG2 cells treated with lipoic acid only, we found an increase in genes involved in mitochondrial fusion and a decrease in those involved in mitochondrial fission (Appendix A). The addition of lipoic acid to HepG2 cells following the exposure to PA:OA reversed the effect brought about by PA:OA. In particular, compared to PA:OA-treated cells, lipoic acid induced a significant increase in the gene expressions of MFN1 (1 μM = 1.46 ± 0.3; 5 μM = 1.32 ± 0.2) and MFN2 (1 μM = 1.19 ± 0.03; 5 μM = 1.42 ± 0.3) (Figure 6A,B) and a significant decrease in the gene expression of FIS1 (1 μM = 0.55 ± 0.07; 5 μM = 0.49 ± 0.04) (Figure 6C). The expressions of genes involved in mitochondrial biogenesis showed that PA:OA treatment causes a significant decrease in the expression of transcription factor A, mitochondrial (TFAM) (0.26 ± 0.035 vs. 1.0 ± 0.09 control), PPARG coactivator 1 alpha (PGC1a) (0.30 ± 0.02 vs. 1.0 ± 0.3 control), and sirtuin 1 (SIRT1) (0.19 ± 0.06 vs. 1.0 ± 0.09 control), (Figure 6D–F) and that this effect was reverted by lipoic acid treatment at both concentrations used (1 and 5 µM) (Figure 6D–F).

These results were further confirmed by Western blot analysis showing that PA:OA treatment resulted in a significant decrease in phosphorylation of DRP1 protein (Figure 7A,B) and a significant increase in SIRT1 (Figure 7D,E) and PGC1alpha (Figure 7D,F) protein levels compared to control cells. Consistently with previous results, the effect of PA:OA were reverted by lipoic acid treatment (Figure 7A–F). Interestingly, lipoic acid treatment resulted in a significant increase in OPA1 protein expression when compared to PA:OA or control cells (Figure 7A,C).

### 2.4. Lipoic Acid Restores the Effect of PA:OA on Autophagy

Our data on autophagy show that palmitic acid/oleic acid determined a significant increase in the percentage of autophagic cells (22.13 ± 1.1) compared to control cells (10.5 ± 0.5) (Figure 8A,B) and that this effect was significantly reduced by lipoic acid, with a more marked effect at the concentration of 1 µM compared (2.87 ± 0.6) to the 5 µM concentration (15.9 ± 1.4). Indeed, both concentrations induced a significant reduction in percentage of autophagic cells compared to PA:OA-treated HepG2 cells (Figure 8A,B) and the 1 µM concentration was able to decrease autophagy even in comparison to untreated cells (Appendix A).

### 2.5. α-Lipoic Acid Restores Energy Metabolism of Steatotic HepG2 Cells

To assess the impact of palmitic acid/oleic acid on cellular energy metabolism, we further analyzed the endogenous metabolic profiles of control, PA:OA-, and lipoic-acid-treated cells. In Figure 8, we report all metabolite changes revealed by the high-performance liquid chromatography (HPLC) analysis of cell extracts. Consistently, our results showed that PA:OA significantly decreased ATP content from 5.56 ± 0.5 nmol/10^6^ cells in control cells to 1.47 ± 0.3 nmol/10^6^ cells in PA:OA-treated cells (Figure 9A). Furthermore (Figure 9A), both lipoic acid concentrations induced a significant increase in ATP content (1 µM = 3.91 ± 0.3 nmol/10^6^ cells; 5 µM = 4.28 ± 0.9 nmol/10^6^ cells, *p* < 0.05 compared to PA:OA-treated cells), clearly indicating that lipoic acid is able to restore the main mitochondrial function, i.e., ATP production to support the cell energy demand. This is confirmed by the ATP:ADP ratio (measuring the mitochondrial phosphorylating capacity) showing a dramatic decrease in PA:OA-treated cells (1.02 ± 0.3 vs. 4.02 ± 0.9 control cells) and a substantial recovery by lipoic acid (ATP:ADP: 1 µM = 3.54 ± 0.3; 5 µM = 3.9 ± 1.2; Figure 9B). The impairment of mitochondrial functions with an increase in glycolytic flux following PA:OA treatment is evidenced by the significantly decreased NAD+ concentration (Figure 9D) and the significantly increased NADH concentration (Figure 9E) (*p* < 0.05 compared to control cells). Also in this case, the effects were significantly reverted by lipoic acid (NAD+: 1 µM = 6.19 ± 1.5; 5 µM = 8.56 ± 0.7; NADH: 1 µM = 0.34 ± 0.04; 5 µM = 0.27 ± 0.02; Figure 9D,E). Changes of the metabolic pathways for energy supply after PA:OA without and with subsequent lipoic acid addition are confirmed by the remarkable changes occurring to the NAD+:NADH ratio (Figure 9F). Additionally, PA:OA causes a significant decrease in NADP+ concentration (0.25 ± 0.01 vs. 0.33 ± 0.02 control cells; Figure 9G) and a more relevant decrease in NADPH concentration (Figure 9H), with a tendency to produce an overall increase in the NADP+/NADPH ratio (Figure 9I). Treatment with lipoic acid after cell challenging with the lipotoxic compounds causes a significant increase in the concentration of both NADP+ and NADPH (NADP+: 1 µM = 0.36 ± 0.008; 5 µM = 0.32 ± 0.02; NADPH:1 µM = 0.19 ± 0.04; 5 µM = 0.22 ± 0.04; Figure 9G,H) compared to PA:OA-treated cells (NADP+: 0.25 ± 0.01; NADPH:0.09 ± 0.01). No significant differences were found when comparing these data to untreated cells (Appendix A), suggesting that under normal conditions, it has no effect on the levels of biosynthetic nicotinic coenzymes (NADP+ and NADPH). Finally, using a Seahorse analyzer, determinants of glycolysis were examined in living cells through the glycolytic rate assay (Figure 9J,K). Our data showed an increase in basal glycolysis in PA:OA-treated HepG2 cells compared with control cells and that treatment with lipoic acid was able to counteract this effect (Figure 9J). Furthermore, OCR was also measured, showing that in HepG2 cells, treatment with PA:OA resulted in a significant decrease in OCR compared to control cells, while α-lipoic acid (at both concentrations used) resulted in a significant increase in OCR compared to PA:OA-treated cells (Figure 9K).

## 3. Discussion

NAFLD is a complex metabolic condition characterized by oxidative stress, fibrosis, and insulin resistance [5]. The complex pathophysiology of this condition involves mitochondrial dynamics, which appears to play a major role [8]. Mitochondrial dynamics refer to the processes governing the structure, function, and distribution of mitochondria within cells. Mitochondria are crucial for cellular energy metabolism and participate in various essential cellular functions, including ATP generation [25], calcium homeostasis [26], ROS production [27], and mitophagy [28]. Growing evidence suggests that alterations in mitochondrial dynamics contribute to the development and progression of steatosis. Changes in mitochondrial morphology, fusion, fission, and distribution have been found to affect lipid metabolism and contribute to the accumulation of fat in liver cells [6,29].

Previous studies have demonstrated that treatment with the mitochondrial fusion protein mitofusin-2 improves mitochondrial function and reduces hepatic steatosis in mouse models of NAFLD [7]. Impaired mitochondrial fusion and reduced fusion activity can lead to fragmented mitochondria, which are less efficient in generating ATP and can result in metabolic dysfunction. Similarly, mitochondrial fission, the process by which mitochondria divide into smaller units, has also been implicated in the development of steatosis [9]. Our results indicate a significant decrease in mitochondrial polarization and a notable increase in mitochondrial fragmentation in HepG2 cells treated with palmitic acid and oleic acid (PA:OA), suggesting that this treatment induces mitochondrial dysfunction, emphasizing the key role of mitochondria in the pathogenesis of steatosis.

Mitochondrial dynamics are regulated by various proteins and signaling pathways, including dynamin-related protein 1 (DRP1), mitofusin 1 and 2 (MFN1/2), and optic atrophy protein 1 (OPA1) [10,30]. Dysregulation of these proteins and their associated pathways can disrupt mitochondrial dynamics and contribute to the development of steatosis. Our results demonstrate that PA:OA treatment significantly decreases the mRNA expression levels of genes involved in mitochondrial fusion (MFN1 and MFN2) and increases the mRNA expression level of FIS1, which is involved in mitochondrial fission. Moreover, mitochondrial quality control mechanisms involve the regulation of several processes such as biogenesis, dynamics, and mitophagy. Failure of these quality control processes leads to mitochondrial dysfunction [11]. Mitochondrial biogenesis requires coordinated expression of nuclear and mitochondrial genes. Peroxisome proliferator-activated receptor gamma coactivator-1α (PGC1α), a key transcriptional activator and master regulator of mitochondrial biogenesis, plays a crucial role in this process by activating other transcription factors involved in nuclear and mitochondrial gene expression, such as TFAM and SIRT1 [12,13]. Activation of PGC1α leads to an increase in mitochondrial mass and substrate oxidation. Interestingly, our data reveal that PA:OA treatment significantly reduces the expression of genes involved in mitochondrial biogenesis (PGC1α, TFAM, and SIRT1). These results were further confirmed by protein expression analysis. In particular, we observed a significant decrease in pDRP1 at Ser637, which is shown to be associated with fission inhibition [31].

Mitochondrial biogenesis is a physiological response to increased energy demand, resulting in elevated AMP: ADP:ATP and NAD+: NADH ratios [15,16]. PGC1α activation can be triggered by elevated AMP levels mediated by AMP-activated protein kinase (AMPK) and increased NAD+ levels mediated by the Sirtuin-1 pathway [17,18]. Our findings align with our targeted metabolomic analysis, showing that induction of steatosis in HepG2 cells significantly reduces ATP, NAD+, and NADP+ content, as well as the oxygen consumption rate (OCR), clearly indicating impaired mitochondrial function.

In addition, autophagy can also affect liver injury during NAFLD [19]. Impairment of autophagy and mitochondrial biogenesis plays a significant role in hepatocyte injury in fatty liver diseases. Our results indicate that PA:OA treatment also leads to a significant increase in the percentage of autophagic cells, suggesting that the degradative process of autophagy is activated to remove damaged and dysfunctional mitochondria for regeneration and energy preservation.

Notably, our results demonstrate that lipoic acid can revert the effects of PA:OA treatment on mitochondrial depolarization and fragmentation, restoring these parameters to control levels. This suggests that lipoic acid has a protective role in mitochondrial dynamics. Additionally, lipoic acid exhibits antioxidant properties that protect mitochondria from oxidative stress, which impairs mitochondrial function and dynamics, including fusion. Consistently, our data show that lipoic acid significantly increases the expression of MFN1 and MFN2, while decreasing FIS1 gene expression. Moreover, lipoic acid reduces the percentage of autophagic cells following PA:OA treatment, indicating its potential to alleviate mitochondrial damage by upregulating mitochondrial fusion expression and inhibiting mitochondrial fission.

Interestingly, several studies reported that lypotoxicity induced by palmitate enhances glycolysis metabolism in hepatic cells leading to cell damage [32]. Our data also showed an increase in the glycolytic rate after PA:OA treatment; this effect was reverted by lipoic acid.

Several studies have shown that lipoic acid can sustain glucose homeostasis and prevent the development of NAFLD in high-fat-diet-fed mice [23]. The possible explanation of this mechanism of action of lipoic acid is that the effect may be dependent on the fact that the lipoic acid used for the experiments is the oxidized form rather than the reduced form. Therefore, according to the results presented, the effect may be dependent on its ability to improve mitochondrial function rather than its direct antioxidant properties.

However, the present in vitro study is limited, considering that the complexity of disease may not mimic the in vivo data, and therefore, further animal and human studies are now required to elucidate whether this effect is dependent on lipoic acid’s beneficial impact on mitochondria and glucose metabolism or a possible direct effect on lipid metabolism.

## 4. Materials and Methods

### 4.1. Cell Culture and Pharmacological Treatments

HepG2 cells (American Type Culture Collection, Manassas, VA, USA) were a kind gift from Prof. Maurizio Parola of the University of Turin. Briefly, low-passage cells were grown in DMEM (Sigma-Aldrich, Milan, Italy) supplemented with 10% FBS (EuroClone, Milan, Italy), 100 U/mL penicillin (Life Technologies, Milan, Italy), and 100 µg/mL streptomycin (Life Technologies) at 37 °C in a humidified incubator in an atmosphere of 95% air and 5% CO_2_. Upon reaching 80–90% confluency, to induce steatosis, HepG2 cells were pretreated for 24 h as follows: HepG2 + vehicle (bovine serum albumin (BSA) 5%); HepG2 + PA:OA (BSA 5% + palmitic acid 250 µM and oleic acid 500 µM). Following 24 h from PA:OA treatment, cells were treated as follows: HepG2 + PA:OA + α-lipoic acid (BSA 5% + palmitic acid 250 µM and oleic acid 500 µM + α-lipoic acid 1–5 µM).

The choice of mixture (PA:OA—250 µM:500 µM) is based on a previous study that showed that the FFA mixture containing a low proportion of palmitic acid (palmitate/oleate 1:2 ratio) is associated with minor toxic and apoptotic effects, thus representing a cellular model of steatosis [33]. The choice of lipoic acid concentration is based on a dose-response curve, which showed that the concentration of 1 and 5 µM is not toxic to cells, and furthermore, it is in a clinically relevant range.

### 4.2. Real-Time Monitoring of Cell Viability

The xCELLigence experiments were performed using a real-time cell analysis (RTCA) dual plate (DP) instrument according to the manufacturers’ instructions (Roche Applied Science, Mannheim, Germany and ACEA Biosciences, San Diego, CA, USA). The RTCA DP instrument comprises three major components: (i) the RTCA DP analyzer, placed inside a humidified incubator maintained at 37 °C and 5% CO_2_; (ii) RTCA control unit with pre-installed RTCA software (Software Version 1.2, Version November 2009); and (iii) E-Plate 16 for the proliferation assay. The E-Plate 16 is a single-use 16-well cell culture plate with bottom surfaces covered by microelectrode sensors (well surface area of 0.2 cm^2^; maximum volume of 243 ± 5 μL). The real-time changes were expressed as a cell index defined as (Rn-Rb)/15, where Rb is the background impedance and Rn is the impedance of the well with cells. Negative control groups were tested in each experiment. Before seeding the cells, background impedance was measured after the 30 min incubation period at room temperature. After seeding 5000 cells in each well, the plate was incubated at room temperature for 30 min; then, the cells were treated with PA:OA and lipoic acid and automatically monitored every 15 min for 72 h. The optimal cell number was determined in a preliminary series of experiments to obtain a significant cell index value and constant cell growth during the entire duration of the experiment.

### 4.3. Oil Red O Staining

Lipid accumulation in HepG2 cells was assessed by Oil Red O staining. In brief, cells were fixed in 4% paraformaldehyde for 30 min, followed by staining with Oil Red O for 30 min. Next, fixed cells were gently washed with isopropanol and, then, washed three times with distilled water. Results were determined using Harmony high-content imaging and analysis software (PerkinElmer, Waltham, MA, USA) following cell segmentation to determine the percentage of steatosis and percentage of macro-steatosis.

### 4.4. Evaluation of Mitochondrial Fragmentation

Mitochondria were stained with 200 nM MitoTracker Red CMX Ros probe (Thermo Fisher Scientific, Milan, Italy) for 30 min at 37 °C, according to the manufacturer’s instructions. Briefly, the cells were seeded in a 96-well multiplate (Cell Carrier Ultra) at a density of 5 × 10^3^ cells. HepG2 cells were pretreated with PA:OA, and after 24 h, cells were treated with the dye for 30 min at 37 °C. Cells were washed 3 times in phosphate-buffered saline (PBS) to remove the unbound probe. Nuclei were stained by NucBlue (two drops per mL) (Thermo Fisher Scientific, Milan, Italy) for 15 min at 37 °C, according to the manufacturer’s instructions, and washed 3 times in phosphate-buffered saline (PBS). Finally, cells were treated with 1–5 µM of α-lipoic acid. For image acquisition, we used Operetta (Perkinelmer, MA, USA), where cells were maintained at 37 °C, and images were captured every 12 h, up to 48 h after treatment. Data collected were analyzed by Harmony software (Version number 4.9; Perkinelmer, MA, USA).

### 4.5. Evaluation of Mitochondrial Membrane Potential (Δψ)

Mitochondrial ∆ψ was assessed in living cells using the fluorescent intensity of the JC-1 probe (10 μg/mL) [32] at emission wavelengths of 585 nm (red fluorescence = high value of ∆ψ) and 527 nm (green fluorescence = low value of ∆ψ). JC-1 permeates the mitochondria as a function of ∆ψ, giving a red fluorescence when the mitochondrial membrane potential is high and shifting into green fluorescence when the mitochondrial membrane potential is decreased.

Briefly, the cells were seeded in a 96-well multiplate (Cell Carrier Ultra) at a density of 5 × 10^3^ cells. After 24 h, the cells were pretreated with PA:OA to induce steatosis and, in the following 24 h, treated with α-lipoic acid and incubated with media containing dye for 1 h. Then, the cells were washed and read in confocal conditions using the 20× long WD objective by a high-content screening (HCS) analysis system (PerkinElmer Operetta High-Content Imaging System) for 48 h.

### 4.6. Real-Time PCR for Gene Expression Analysis

HepG2 the cells were pretreated with PA:OA to induce steatosis and, in the following 6 h, treated with α-lipoic acid. RNA was extracted using Trizol^®^ reagent (Invitrogen, Carlsbad, CA, USA) [34]. First-strand complementary DNA (cDNA) was then synthesized with a reverse transcription reagent from Applied Biosystems (Foster City, CA, USA). Quantitative real-time PCR (qRT-PCR) was performed in a StepOne Fast Real-Time PCR System (Applied Biosystems) using the SYBR Green PCR MasterMix (Life Technologies, Monza, Italy). The specific PCR products were detected with SYBR green fluorescence. The relative messenger RNA (mRNA) expression level was calculated by the threshold cycle (Ct) value of each PCR product and normalized with that of actin using a comparative 2−DDCt method. The sequences of the primers used are presented in Table 3.

### 4.7. Western Blot Analysis

Briefly, for Western blot analysis, 50 μg of proteins were loaded onto a 12% polyacrylamide gel Mini-PROTEAN^®^ TGXTM (BIO-RAD, Milan, Italy). Electro-transfer to nitrocellulose membrane was obtained through Trans-Blot^®^ TurboTM (BIO-RAD) using a Trans-Blot^®^ SE Semi-Dry Transfer Cell (BIO-RAD). Membranes were blocked in Odyssey Blocking Buffer (Licor, Milan, Italy), according to the manufacturer’s protocol. After blocking, membranes were washed three times in PBS for 5 min and incubated with primary antibodies against SIRT1 (ab189494), DRP1 (ab184247), pDRP1 S637 (ab193216), OPA1 (ab157457), and PGC1alpha (ab77210) (ABCAM) overnight at 4 °C. The next day, membranes were washed three times in PBS for 5 min and incubated with anti-rabbit IRDye700CW secondary antibodies (1:5000) in PBS/0.5% Tween-20 for 1 h at room temperature. All the antibodies were diluted in Odyssey Blocking Buffer. The obtained blots were visualized by an Odyssey Infrared Imaging Scanner (Licor, Milan, Italy). Densitometric analysis was used for protein level quantification, normalizing data to protein levels of β-actin.

### 4.8. Immunocytochemical Analysis

After washing with PBS, cells were treated with a MitoTracker Red CMXRos probe (Thermo Fisher Scientific, Milan, Italy) for 30 min at 37 °C, and it was removed after 30 min. At this stage, cells were washed 3 times in phosphate-buffered saline (PBS) to remove the unbound probe. After washing with PBS, cells were fixed in 4% paraformaldehyde (category no. 1004968350 Sigma-Aldrich, Milan, Italy) for 20 min at room temperature. Subsequently, cells were incubated with human LC3A antibody FITC-conjugated (NB100-2331F, Novus Biologicals) at a dilution of 1:200 overnight at 4 °C. The next day, cells were washed three times in PBS for 5 min and the nuclei were stained by NucBlue (two drops per mL) (Thermo Fisher Scientific, Milan, Italy) for 15 min at 37 °C, according to the manufacturer’s instructions [35]. The fluorescent images were obtained using Operetta (PerkinElmer, MA, USA).

### 4.9. Metabolomic Profile by HPLC Analysis

At the end of incubation under the different experimental conditions (controls, PA:O, PA:O + lipoic acid), cells (3 × 10^6^, *n* = 6 replicates) were washed twice with large volumes of ice-cold 10 mM PBS at pH 7.4 and then centrifuged at 1860× *g* for 5 min at 4 °C. Cell pellets were vigorously mixed with 1 mL of ice-cold, nitrogen-saturated, deproteinizing solution (10 mM KH_2_PO_4_ + HPLC-grade CH_3_CN, pH 7.4, 1:3 *v*:*v*) and centrifuged at 20,690× *g* for 10 min at 4 °C. The supernatants were mixed with two volumes of HPLC-grade chloroform and centrifuged (20,690× *g* for 10 min at 4 °C), and the upper aqueous phase was recovered and used for the HPLC analysis of metabolites. The simultaneous separation and quantification of adenine nucleotides (ATP, ADP, and AMP) and nicotinic coenzymes (NAD+, NADH, NADP+, NADPH) in the protein-free cell extracts were carried out using established HPLC methods [36,37,38].

### 4.10. Quantification of Cellular Glycolytic Rate Using Agilent Seahorse XF Technology

Mitochondrial respiration was measured using Agilent’s cell glycolytic rate assay kit (Agilent Technologies (Santa Clara, CA, USA), PN: 103344-100) and a Seahorse XFe24 analyzer. Drugs that were injected into the assay were prepared according to manufacturer’s instructions (2.0 µM oligomycin, 1.0 µM FCCP, and 0.5 µM rotenone/antimycin A). Oxygen consumption rates (OCR), extracellular acidification rate (ECAR), and proton efflux rate (PER) were determined by sequential measurement cycles consisting of a 30 s mixing time followed by a two-minute wait time and then a three-minute measurement period (three measurements following each reagent addition). Reagents were added to the SeahorseXFe24 FluxPak in dilutions according to the manufacturer’s recommendation (2.0 μM oligomycin, 1.0 μM carbonyl cyanide-p-trifluoromethox-yphenyl-hydrazon (FCCP), 0.5 μM rotenone/antimycin A per well). The first three measurements of OCR and PER occur prior to the addition of mitochondrial reagents and indicates basal respiration. Oligomycin is a complex V inhibitor, and OCR, following this addition, indicates ATP-linked respiration (subtraction of baseline OCR) and proton leak respiration (subtraction of non-mitochondrial respiration). FCCP is a protonophore, and adding it will collapse the inner membrane gradient and push the electron transport chair to the maximal rate. Inhibition of complex III and I is achieved through the addition of antimycin A and rotenone, respectively, and will terminate electron transport chain function and demonstrate non-mitochondrial respiration. Mitochondrial reserve is calculated by subtraction of basal respiration from maximal respiration. The XF reports of the glycolytic rate assay data were recorded in Agilent’s Wave software (Wave Controller Software 2.6), and subsequently, data were exported to Excel and Prism for further analysis and visualization.

### 4.11. Statistical Analysis

Statistical analysis was performed using GraphPad Prism software, version 9.0 (GraphPad Software Inc., San Diego, CA, USA, RRID: rid_000081). For comparison of *n* ≥ 3 groups, one-way analysis of variance (ANOVA) with the Holm–Sidak post hoc test for multiple comparisons was used. Data are expressed as mean ± SD, unless otherwise stated, and *p*-values < 0.05 were considered statistically significant.

## 5. Conclusions

In conclusion, there is a significant association between steatosis and mitochondrial dynamics. Alterations in mitochondrial fusion, fission, and distribution can influence the metabolic function of liver cells and contribute to intracellular lipid accumulation. While more research is needed to specifically investigate the effects of lipoic acid on mitochondrial fission and fusion, its impact on mitochondrial health, antioxidant activity, and energy metabolism suggests that it may play a role in supporting optimal mitochondrial dynamics, including fission. Understanding the underlying mechanisms involved in these processes may provide insights into the development of therapeutic strategies for the treatment of steatosis.

## Figures and Tables

**Figure 1 ijms-24-14491-f001:**
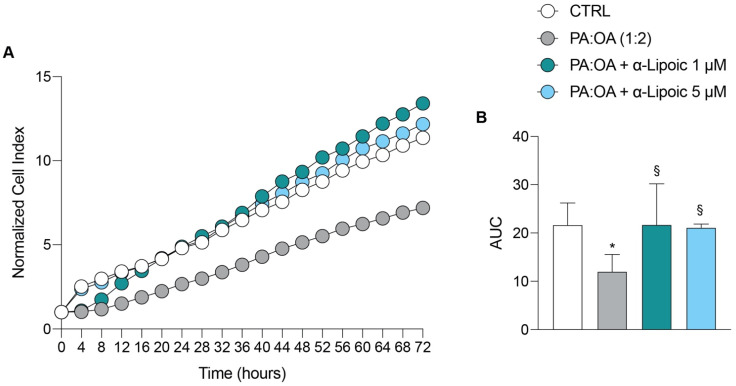
Real-time cell proliferation of HepG2 cells after palmitic acid/oleic acid (PA:OA) treatment and α-Lipoic acid treatment. (**A**) Normalized cell index. (**B**) Area under curve. The cell index values were normalized at the time of pharmacological treatments in order to obtain a normalized cell index. Each line expresses the average of four different experiments. * vs. CTRL (* *p* < 0.05); ^§^ vs. PA:OA (^§^
*p* < 0.05).

**Figure 2 ijms-24-14491-f002:**
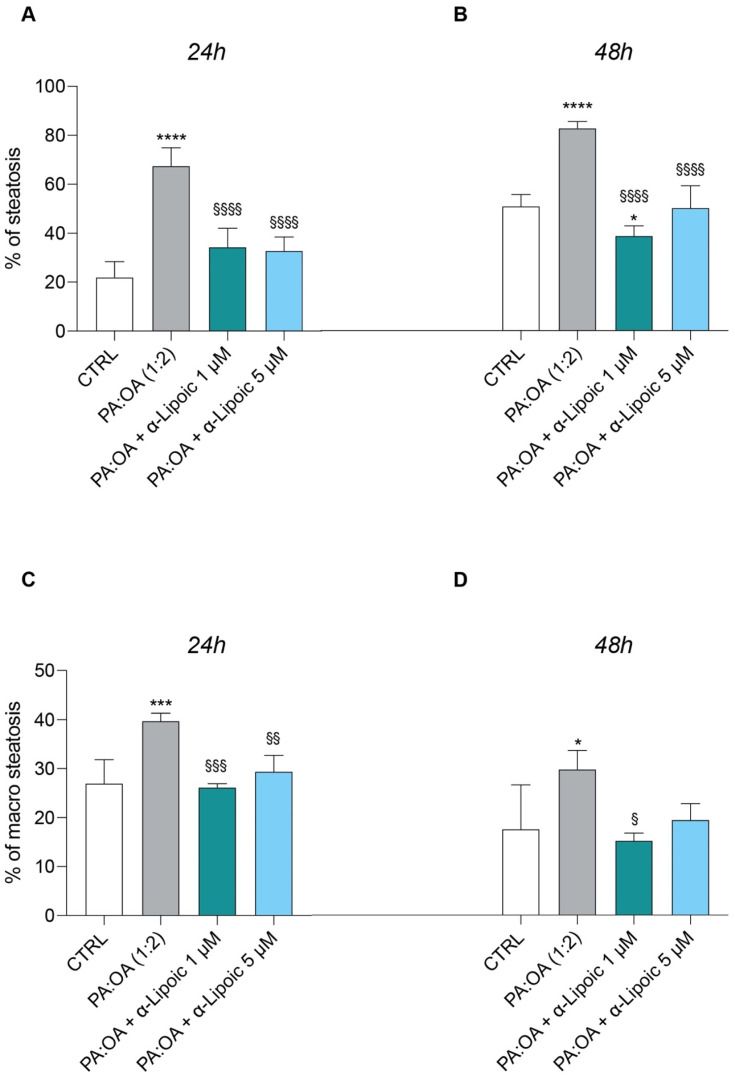
Computerized analysis of % of steatosis and % of macro steatosis after acid/oleic acid (PA:OA) treatment and α-lipoic acid treatment. (**A**,**C**) 24 h and (**B**,**D**) 48 h of treatment. The % of macro-steatosis was calculated within the cell population with steatosis. Values represent the mean ± SD of experiments performed in quadruplicate. * vs. CTRL (* *p* < 0.05, *** *p* < 0.001, **** *p* < 0.0001); ^§^ vs. PA:OA (^§^
*p* < 0.05, ^§§^
*p* < 0.01, ^§§§^
*p* < 0.001, ^§§§§^
*p* < 0.0001).

**Figure 3 ijms-24-14491-f003:**
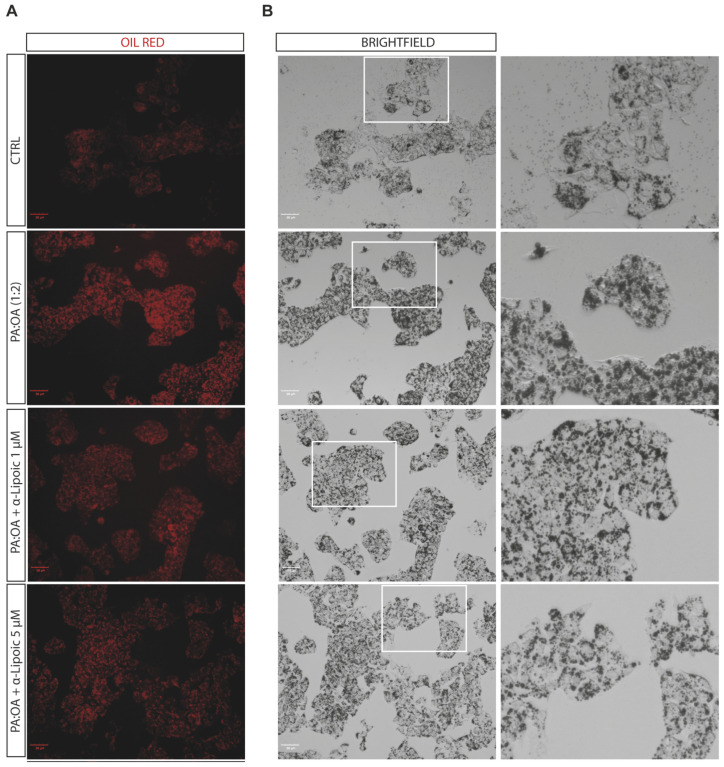
Computerized analysis of the Oil Red in the control versus PA:OA and α-lipoic acid after 24 h of treatment. (**A**) oil red staining; (**B**) Brightfield. The figures presented are representative of at least three independent experiments. Scale bars in (**A**,**B**) 50 μm.

**Figure 4 ijms-24-14491-f004:**
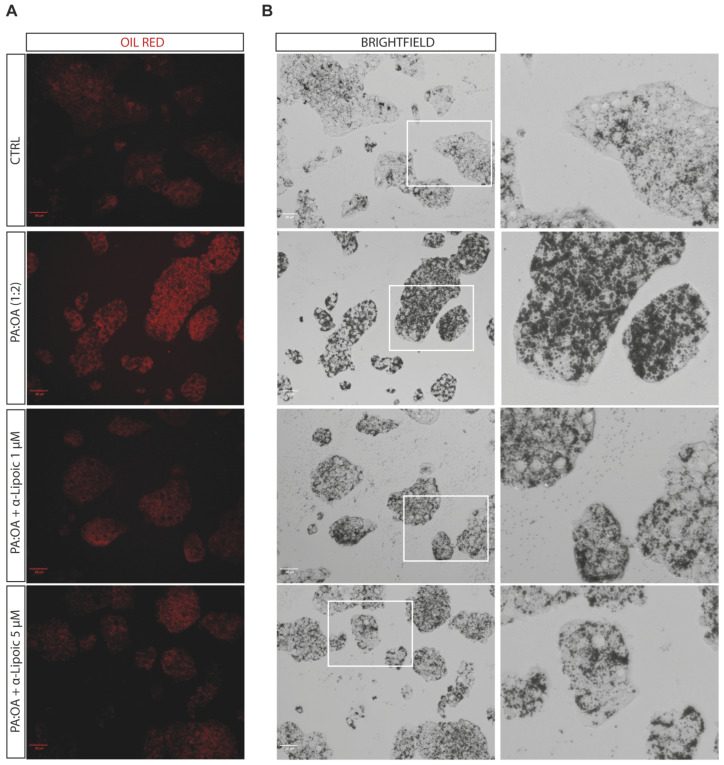
Computerized analysis of the Oil Red in the control versus PA:OA and α-lipoic acid after 48 h of treatment. (**A**) oil red staining; (**B**) Brightfield. The figures presented are representative of at least three independent experiments. Scale bars in (**A**,**B**) 50 μm.

**Figure 5 ijms-24-14491-f005:**
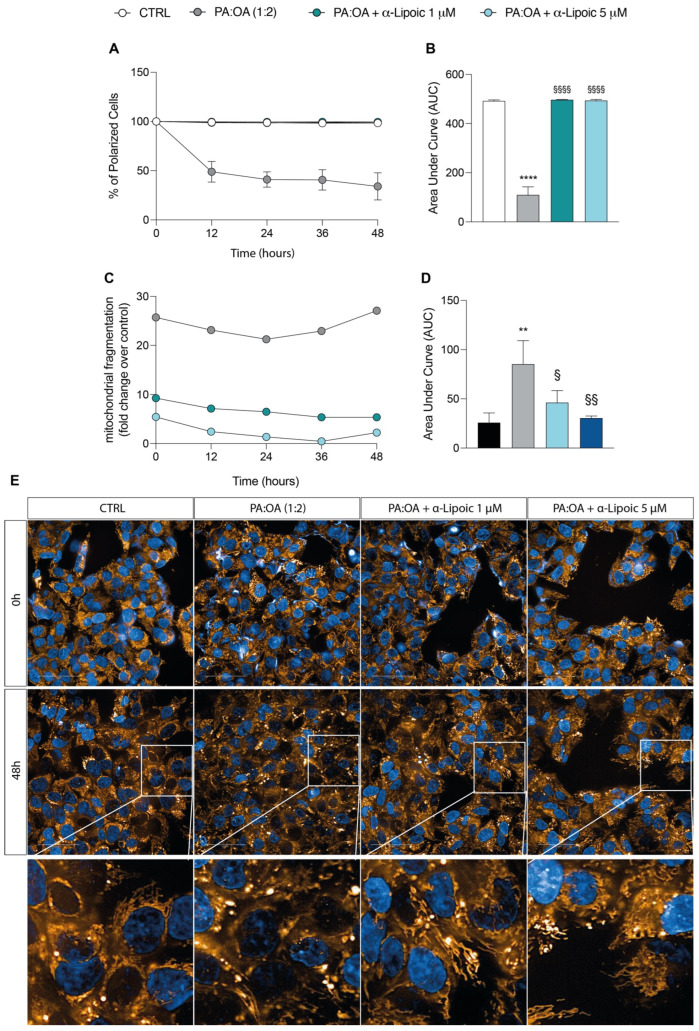
Computerized analysis of JC1 and MitoTracker Red after palmitic acid/oleic acid (PA:OA) and α-lipoic acid. (**A**) % of polarized cells. (**B**) Area under curve. (**C**) Mitochondrial fragmentation. (**D**) Area under curve. * vs. CTRL (** *p* < 0.01, **** *p* < 0.0001); ^§^ vs. PA:OA (^§^
*p* < 0.05, ^§§^
*p* < 0.01, ^§§§§^
*p* < 0.0001). (**E**) Computerized analysis of the MitoTracker fluorescence intensity on the control versus PA:OA and α-lipoic acid after 48 h of treatment. The figures presented are representative of at least three independent experiments. Scale bars in (**E**) 50 μm.

**Figure 6 ijms-24-14491-f006:**
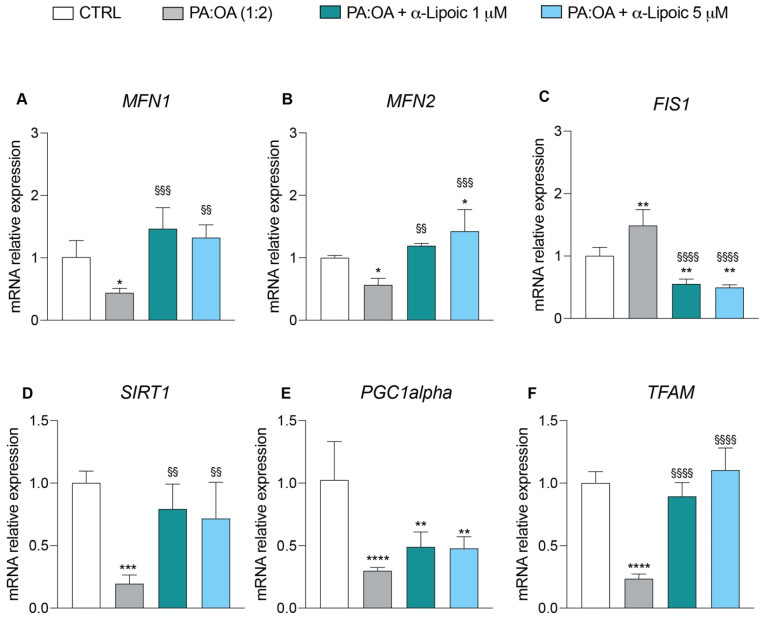
Real-time PCR analysis of genes involved in mitochondrial dynamics and biogenesis. mRNA expression levels of (**A**) MFN1, (**B**) MFN2, (**C**) FIS1, (**D**) SIRT1, (**E**) PGC1α, and (**F**) TFAM after 6 h of treatment. Values represent the mean ± SD of experiments performed in quadruplicate. * vs. CTRL (* *p* < 0.05, ** *p* < 0.01, *** *p* < 0.001, **** *p* < 0.0001); ^§^ vs. PA:OA (^§§^
*p* < 0.01, ^§§§^
*p* < 0.001, ^§§§§^
*p* < 0.0001).

**Figure 7 ijms-24-14491-f007:**
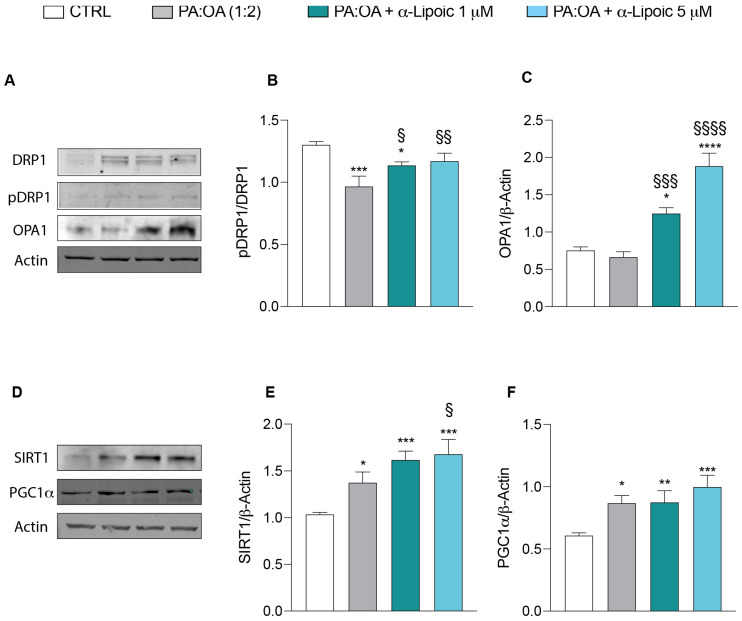
Western blot analysis of proteins involved in mitochondrial dynamics and biogenesis. Protein expression levels of (**A**,**B**) pDRP1 and DRP1, (**A**–**C**) OPA1, (**D**,**E**) SIRT1, and (**D**–**F**) PGC1α after 24 h of treatment. Values represent the mean ± SD of experiments performed in quadruplicate. * vs. CTRL (* *p* < 0.05, ** *p* < 0.01, *** *p* < 0.001, **** *p* < 0.0001); ^§^ vs. PA:OA (^§^
*p* < 0.05, ^§§^
*p* < 0.01, ^§§§^
*p* < 0.001, ^§§§§^
*p* < 0.0001).

**Figure 8 ijms-24-14491-f008:**
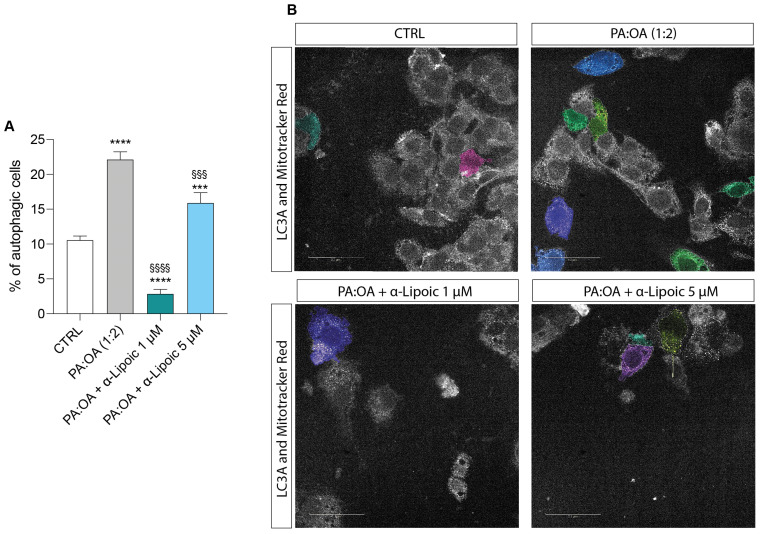
Computerized analysis of LC3A and % of autophagic cells. * vs. CTRL (*** *p* < 0.001, **** *p* < 0.0001); ^§^ vs. PA:OA (^§§§^
*p* < 0.001, ^§§§§^
*p* < 0.0001). (**A**) quantification of % of autophagic cells; (**B**) Representative images of autogaphy-positive cells. The figures presented are representative of at least three independent experiments. Scale bars in (**B**) 50 μm.

**Figure 9 ijms-24-14491-f009:**
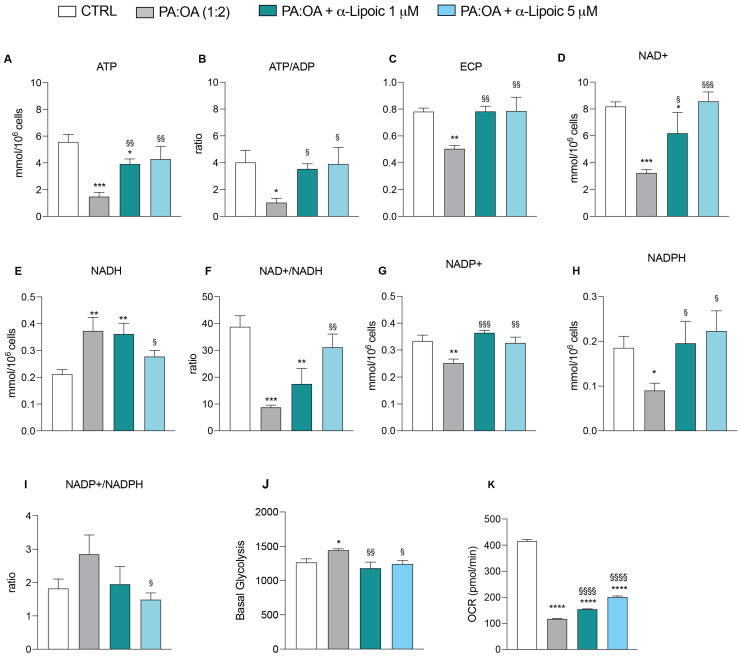
HPLC analysis and Seahorse analysis. Effect of PA:OA and α-lipoic acid treatment on levels of major classes of metabolites detected by HPLC (**A**–**I**) and effect of PA:OA and α-lipoic acid treatment on basal glycolysis and oxygen consumption rate (OCR) detected by Seahorse (**J**,**K**). Values represent the mean ± SD of experiments performed in quadruplicate. * vs. CTRL (* *p* < 0.05, ** *p* < 0.001, *** *p* < 0.001, **** *p* < 0.0001); ^§^ vs. PA:OA (^§^
*p* < 0.05, ^§§^
*p* < 0.01, ^§§§^
*p* < 0.001, ^§§§§^
*p* < 0.0001).

**Table 1 ijms-24-14491-t001:** Normalized cell index raw data values.

	12 h	24 h	36 h	48 h	72 h	AUC
CTRL	3.41 ± 0.9	4.81 ± 1.06	6.47 ± 1.6	8.26 ± 1.9	11.35 ± 2.7	21.64 ± 4.5
PA:OA (1:2)	1.50 ± 0.8 *	2.67 ± 0.8 *	3.81 ± 1.1	5.14 ± 1.4	7.18 ± 2.04	11.98 ± 3.5
PA:OA + α-Lipoic 1 μM	2.71 ± 1.7	4.88 ± 2.3 ^§^	6.88 ± 2.6	9.33 ± 3.4	13.40 ± 4.3 ^§^	21.69 ± 8.4
PA:OA + α-Lipoic 5 μM	3.34 ± 0.32	4.88 ± 0.1 ^§^	6.74 ± 0.1	8.72 ± 0.5	12.17 ± 1.0	21.08 ± 0.7

* vs. CTRL (* *p* < 0.05), ^§^ vs. PA:OA (^§^
*p* < 0.05).

**Table 2 ijms-24-14491-t002:** Cellomics analysis.

		CTRL	PA:OA (1:2)	PA:OA + α-lipoico 1 µM	PA:OA + α-lipoico 5 µM
Steatotic cells—Total Lipid droplets Area	12 h	178.5 ± 14.3	224.95 ±7.4 ***	176.6 ± 2.5 ^§§§^	188.3 ±11.6 ^§§^
	48 h	144.7 ± 31.2	190.1 ± 10.8 *	140.2 ± 5.4 ^§^	154.6 ± 9.0
Steatotic cells—Number of Lipid droplets	12 h	6.99 ± 0.48	6.7 ± 0.04	6.2 ± 0.1	6.5 ± 0.3
	48 h	6.2 ± 1.4	5.7 ± 0.3	5.6 ± 0.1	5.9 ± 0.2
Steatotic cells—Number of Lipid droplets per Area of Cytoplasm	12 h	0.020 ± 0.0006	0.017 ± 0.0002 **	0.019 ± 0.0006 ^§§^	0.019 ± 0.0003 ^§§^
	48 h	0.019 ± 0.0009	0.017 ± 0.0003 *	0.018 ± 0.0003	0.019 ± 0.0003 ^§§^
Steatotic cells—Lipid droplets Area (µm²)	12 h	63.7 ± 5.1	80.4 ± 2.7 ***	63.1 ± 0.9 ^§§§^	67.2 ± 4.1 ^§§^
	48 h	51.7 ± 11.2	67.9 ± 3.8 *	50.1 ± 1.9 ^§^	55.19 ± 3.2
Total cells—Total Lipid droplets Area	12 h	158.75 ± 8.2	217.2 ± 7.2 ****	164.1 ± 4.3 ^§§§§^	172 ± 11.3 ^§§§§^
	48 h	126.3 ± 18.9	177.9 ±13.1 ***	112.8 ± 2.5 ^§§§^	140.0 ± 7.2 ^§§^
Total cells—Number of Lipid droplets	12 h	6.8 ± 0.2	6.52 ± 0.05	6.03 ± 0.1 **	6.2 ± 0.3 *
	48 h	5.7 ± 1.0	5.4 ± 0.3	5.19 ± 0.07	5.6 ± 0.2
Total cells—Number of Lipid droplets per Area of Cytoplasm	12 h	0.02 ± 0.0006	0.018 ± 0.0003 ****	0.02 ± 0.0007 ^§§^	0.02 ± 0.0003 ^§§§^
	48 h	0.019 ± 0.0006	0.017 ± 0.0003 **	0.018 ± 0.0004	0.019 ± 0.0004 ^§§§^
Total cells—Lipids droplets Area (µm²)	12 h	56.5 ± 2.9	77.68 ± 2.6 ****	58.71 ± 1.5 ^§§§§^	61.5 ± 4.0 ^§§§§^
	48 h	45.2 ± 6.7	63.6 ± 4.6 ***	43.9 ± 0.9 ^§§§^	50.1 ± 2.5 ^§§^
Total cells—Cell Volume (µm³)	12 h	3312.87 ± 1036.7	4039.5 ± 2256.5	2299.37 ± 787.3	3741.5 ± 1731
	48 h	1727 ± 528.8	2014.2 ± 986.2	1585.7 ± 433.3	1903.25 ± 794.1
Total cells—Cell Surface Area (µm²)	12 h	3359.3 ± 473.5	4449.8 ± 2846.1	3414.1 ± 834.9	4444.3 ± 2333.8
	48 h	1726.8 ± 285.8	2162.2 ± 1089.7	1753.7 ± 442.3	2261 ± 1239.4

* vs. CTRL (* *p* < 0.05, ** *p* < 0.01, *** *p* < 0.001, **** *p* < 0.0001); ^§^ vs. PA:OA (^§^
*p* < 0.05, ^§§^
*p* < 0.01, ^§§§^
*p* < 0.001, ^§§§§^
*p* < 0.0001).

**Table 3 ijms-24-14491-t003:** Gene of interest.

Genes	Forward Primer (5′⟶3′)	Reverse Primer (5′⟶3′)	Accessionnumber
*MFN1*	TCGGGAAGATGAGGCAGTTT	TGCCATTATGCTAAGTCTCCG	NM_033540.3
*MFN2*	CGGGAAGGTGAAGCGCAATG	ACCAGGAAGCTGGTACAACG	NM_001127660.2
*OPA1*	GAAAGGAGCTCATCTGTTTGGAGTC	TTCTTCCGGAGAACCAAAATCG	NM_001354663.2
*FIS1*	ACTACCGGCTCAAGGAATACG	CATGCCCACGAGTCCATCTT	NM_016068.3
*PGC1ɑ*	ATGAAGGGTACTTTTCTGCCCC	GGTCTTCACCAACCAGAGCA	NM_001330751.2
*TFAM*	CCGAGGTGGTTTTCATCTGT	AGTCTTCAGCTTTTCCTGCG	NM_003201.3
*SIRT1*	AGGCCACGGATAGGTCCATA	GTGGAGGTATTGTTTCCGGC	NM_012238.5
*β-Actin*	CCTTTGCCGATCCGCCG	AACATGATCTGGGTCATCTTCTCGC	NM_001101.5

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
