# Peer review of "(+)-Lipoic Acid Reduces Lipotoxicity and Regulates Mitochondrial Homeostasis and Energy Balance in an In Vitro Model of Liver Steatosis"

_ijms, 2023, doi:10.3390/ijms241914491_

Round 1
Reviewer 1 Report
The study presented by the authors aims at evaluating the beneficial effects of ALA in an in vitro model of liver steatosis. They use several methodologies to characterize some of the metabolic effects of a palmitic acid/oleic acid mixture (1:2) and the beneficial effects of supplementation of cell culture media with two concentrations of ALA.
Major comments:
i) In the introduction the authors should justify more clearly the choice of the lipidic mixture and of the concentrations of ALA used.
ii) Results presented by the authors show beneficial effects of the presence of ALA in most of the studies performed but fail to address the fact that in most circumstances there is no major difference between the results obtained using 1 mM or 5 mM concentrations. Is it possible that the range of concentrations used was not adequately chosen or can the authors provide a comprehensive explanation on the matter?
iii) The authors provide results using many methodologies and report ALA as a compound capable of reverting the effects of PA:OA mixture. Nevertheless, discussion fails to provide an integrative picture of the whole data set and does not provide possible mechanism(s) of action of ALA. This would greatly benefit the manuscript.
i) The choice for the abbreviation ALA could be confound with Alanine. Is there any other alternative?
ii) The presentation of PA:OA mixture and subsequently 250 mM/500 mM for the respective concentrations. Why not showing the concentrations also with the format 250 mM:500 mM.
iii) Page 10 line 192 – revise; as is makes no sense.
iv) Page 10 line 197: wassignificantly – correct.
v) Page 10 line 209: a-lipoic acid – a needs to be put as a.
vi) Page 11 line 223: what is the word “was” doing on the sentence?
vii) Page 11 line 225: the result for the 1 mM ALA condition referring to NADH is missing!
viii) Page 11 lines 228-230: the result presented does not make sense; if NADP+ decreases and NADPH does not change, how is there a tendency for an increase in the ratio NADP+/NADPH?
ix) Page 11 line 235: “whan” – correct.
x) Page 11 lines 241-244: “However, OCR ….”; was the result found unexpected? If that was not the case why starting the sentence by However?
xi) Page 13 line 291: “AMP: ADP/ATP and NAD+:NADH ratios”. The way the authors refer to ratios should by homogenised throughout the manuscript.
xii) Page 14 line 321: “mitochondrial and glucose metabolism”; what do the authors mean by this? Should it be mitochondria instead of mitochondrial?
Reviewer 2 Report
In the current manuscript, authors described the effects of Lipolic acid on lipid accumulation and mitochondrial dynamics fusion & fission in in vitro model of NAFLD. The complexity of NAFLD pathophysiology and the phenotypic heterogeneity, both genetic and environmental factors are thought to contribute to disease progression and susceptibility. In vitro models are, in many settings, a suitable alternative for studying NAFLD and there is increasing sophistication and ability to recapitulate several hallmarks of the disease. A multi-disciplinary approach to the development of in vitro models has led to solutions far more complex than the traditional two-dimensional (2D) cultures. Examples include liver-on-a-chip platforms and three-dimensional (3D) models. By supporting a better understanding of the disease, these models could be used as reliable drug screening platforms. However, cell line model of NAFLD has several challenges in recapitulating the in vivo studies as the signaling involved with several other cell types inflammatory and other cell types in Liver. As authors used the HepG2 cells treated with fatty acids as model of NAFLD and tested the pharmacological compound on regulating the lipotoxicity and mito dynamics. Based on the results indicated manuscript, Lipoic acid has positive/beneficial effects in prohibiting the lipid accumulation and improved the mitochondrial function and all the results convincing the hypothesis. However, current cell culture data alone has limited the scope the manuscript and I would recommend to submit the manuscript other suitable journal as the has several limitations and understanding the complexity of NAFLD progression and studying in vivo animal model could have been advantage considering the complexity of disease.
Reviewer 3 Report
The article of Lucia Longhitano et al. “(+)-Lipoic Acid Reduces Lipotoxicity and Regulates Mitochondrial Homeostasis and Energy Balance in an In Vitro Model of Liver Steatosis” describes the use of lipoic acid for the treatment of non-alcoholic fatty liver disease. The authors showed that lipoic acid reduces lipotoxicity and improves functional state of mitochondria in model of steatosis (in vitro).
The paper is well presented and well written.
However, there are some comments on submitting the article.
1. “in vitro” … replace with “in vitro” (check all text)
2. line 31…”uM” replace with “µM”
3. Each figure legend should have a title such as figure 3. The authors have figure legends as results.
4. In the description of the results, it is necessary to show by how much % this or that parameter has decreased or increased.
For example, line 88-90…“Our results showed that PA:OA treatment induces a significant decrease in normalized cell index and in the area under the curve (AUC) (11.97 ± 3.5), compared to control cells (21.64 ± 4.5), indicating a reduction in cell viability, as shown in Figure 1A-B and Table 1”.
Our results showed that PA:OA treatment caused a significant decrease in normalized cell index and area under the curve (AUC) by 1.8 times compared to control cells, indicating a decrease in cell viability (Figure 1A-B, Table 1).
The numerical value is in the table and the diagram; they should not be inserted into the text.
Make corrections to the text.
5. lines 192-194 … A reference(s) is required at the end of a sentence;
6. line 197… “wassignificantly” replace with “was significantly”;
7. line 200…”the1µM” replace with “the 1 µM”;
8. line 207… ”3.5É‘-. lipoic acid restore energy metabolism of steatotic HepG2 cells” replace with ”2.5 α -lipoic acid restore energy metabolism of steatotic HepG2 cells”;
9. lines 215, 213, 364 and other ”103 or 106” replace with ”103 or 106” - correct throughout
10. line 441 antimicin A and rotenone, respectively
11. The authors discuss mitophagy and present results with an autophagy marker (LC3). Mitophagy markers are PINK1, Parkin. Did the authors investigate the change in mitophagy (PINK1, Parkin) in their experimental conditions).
12. line 417… “10 mM KH2PO4” replace with “10 mM KH2PO4” and CH3CN - CH3CN (check al text)
13. method 4.8 – references
14. Check punctuation in text
Moderate editing of English language required
Round 2
Reviewer 2 Report
Thanks to authors for the responses. However, there are concerns regarding the results.
In Figure 6, results showing the transcript expression of genes involved in Mitochondrial dynamics and biogenesis and Sirt1. As it is known that there are other major post translationally regulated proteins like DRP-1/p-DRP1 and OPA1 involved in the fission/fusion process. I would suggest authors to show the results in protein expression levels by westernblot including Sirt1 and PGC1a.
In methods, authors need to indicate which confocal microscopic system used for imaging?
Author Response
Q1: In Figure 6, results showing the transcript expression of genes involved in Mitochondrial dynamics and biogenesis and Sirt1. As it is known that there are other major post translationally regulated proteins like DRP-1/p-DRP1 and OPA1 involved in the fission/fusion process. I would suggest authors to show the results in protein expression levels by western blot including Sirt1 and PGC1a.
A1: We would like to thank the reviewer for his valuable comments improving the scientific quality of our manuscript. We performed additional protein expression experiments as requested for the indicated proteins. This set of experiments are consistent with our previous observations and are now presented in new figure 7 showing that PA:OA resulted in a significant decrease of pDRP1/DRP1 and a significant increase of SIRT1 and PGC1alpha following PA:OA treatment. Consistently, lipoic acid reverted these effects. Finally, lipoic acid increased OPA1 expression when compared to PA:OA or control cells. We have now included these changes in material and methods section (lines 437-449 page 16), in the results section (lines 199-210 page 10 and new figure 7) and discussion (lines 312-313 page 13).
Round 3
Reviewer 2 Report
Thank you. as I notified earlier that the current study has limitation that in vivo studies considering fact that the complexity of disease may not mimic with the in vitro data. authors need to include the limitation of study.
Author Response
Q1: Thank you. as I notified earlier that the current study has limitation that in vivo studies considering fact that the complexity of disease may not mimic with the in vitro data. authors need to include the limitation of study.
A1: We would like to thank the reviewer for this valuable comment. We have now further underscored the limit of our study in the discussion (page 14 lines 348-352).